# Grandparents' wealth and the body mass index trajectories of grandchildren

**Ying Huang***

Department of Demography, College of Public Policy, University of Texas at San Antonio, San Antonio, TX, United States of America

* ying.huang@utsa.edu

## Abstract

### Background

The aims of this study are to (i) examine associations between grandparents' wealth and grandchild's initial body mass index (BMI) in early childhood and its subsequent growth patterns, and to (ii) assess whether the associations are similar for white and black children.

### Methods

Data are from the U.S. Panel Study of Income Dynamics (PSID) and its supplemental studies of Child Development Supplement (CDS) and Transition to Adulthood (TA) ($N$ = 2,128). Three-level growth curve models are used to analyze the association between exposure to grandparental wealth in early childhood and grandchildren's BMI growth trajectories, accounting for parental sociodemographic characteristics and maternal BMI levels.

### Results

Children with less grandparental wealth in early childhood have higher initial BMI than children with more grandparental wealth. Further, increases in grandparental wealth in childhood are associated with a slower BMI growth rate. The wealth-body mass index associations are more evident among white children than black children.

### Conclusions

The study reveals a multigenerational social gradient to body mass index. Elevating the wealth levels of the grandparent generation could potentially reduce their grandchildren's obesity risk. The protective role of grandparental wealth seems to be more evident among white families than black families.

## Background

The prevalence and severity of obesity have been increasing in children and adolescents [1,2] Despite intense focus on reducing the childhood obesity epidemic over the past decades, the

**Data Availability Statement:** The data underlying the results presented in the study are available from Institute for Social Research at University of Michigan. https://psidonline.isr.umich.edu/.

**Funding:** The author(s) received no specific funding for this work.

**Competing interests:** NO authors have competing interests.

progress remains unclear [3–5]. Research focusing on social determinants of child obesity suggests that lower socioeconomic status (SES) and the associated behavioral and environmental factors [6–9] are important risk factors for excess body weight and obesity during childhood and adolescence [10, 11]. Abnormal weight gain during critical developmental stages may also cause major physical and psychosocial problems, both in the short- and long-run [12–16]. Prior research using such SES indicators as parental education and income may fail to capture the complexity of the social distribution of (dis)advantages. Grandparental wealth is a valuable indicator that reflects persistent inequalities in socioeconomic resources. However, our understanding of the role of grandparental SES in general, and wealth in particular, in influencing the developmental trajectories of obesity risk among children has only begun to emerge [17, 18].

High levels of body mass index (BMI) and excessive BMI growth during childhood are established obesity risk factors [19–22]. For example, high BMI at school entry is a strong predictor of obesity in adolescence, [23]and infants with high BMI are at higher risk of obesity later in childhood and adulthood compared with other children with normal BMI levels [24, 25]. Adverse SES exposure and under-nutrition during the fetal life and early childhood might lead to an elevated risk for excessive catch-up growth or "early adiposity rebound" [26, 27]. This type of early childhood catch-up growth, in turn, is a well-established predictor of adult obesity [28–30] and other chronic diseases [31, 32]. In contract, affluent family environment is an important protective factor that prevents children from obesity risks [16].

Grandparental wealth is a critical SES marker that can play a pivotal role in influencing their grandchildren's weight status beginning in early childhood. Grandparental wealth is an important financial resource that shapes the opportunities structure that a child faces and the environment that a child lives in. Grandparental wealth may play a role in influencing grandchildren's BMI growth trajectories through the "purchasing" function of wealth. In other words, grandparental wealth may help grandchildren to access material goods and services that mitigate obesity risks. For example, ethnographic research suggests that wealthy grandparents often help their adult parents with home purchases in good neighborhoods or help with education expenses [33–35]. Because residential environments can influence heathy food access as well as the opportunities for physical activities [36–38], children residing in quality neighborhoods often have more playgrounds, sidewalks, and recreational facilities than peers residing in poor neighborhoods [39]. In addition, grandparental wealth may play a role in influencing grandchildren's BMI growth trajectories through the "cultural" function of wealth. For instance, wealthy grandparents may help grandchildren form a healthy lifestyle and diet patterns by providing grandchildren with nutritious food or snacks, by paying for extracurricular sports activities, and by encouraging an active lifestyle. This type of healthy lifestyle, in turn, can lead to a higher level of physical activity participation later in life and play a role in obesity prevention [40, 41].

The influence of grandparental wealth on grandchildren's BMI growth trajectories can also operate through the parent generation. Family wealth is a unique set of resources that can be passed from parents to offspring through inter-vivos transfers (e.g. gifts) and after death in the form of inheritances and bequests [42]. The return on capital arising from bequests or gifts enable parents to purchase resources that may enhance children's access to better nutrition, health, and good housing and neighborhood conditions. All these resources, in turn, could play an important role in preventing children from excessive weight gain and obesity risk [43].

The association between grandparental wealth and children's BMI growth patterns can differ by grandchildren's race and ethnicity. A tremendous wealth gap between non-Hispanic white and black families is well documented, and white children tend to enjoy much more grandparental wealth than their black peers [33, 44, 45]. However, existing research addressing

racial differences in the effects of grandparental wealth on child BMI growth trajectories is limited. It is possible that the impact of grandparental wealth is stronger among blacks than whites. This is because blacks tend to regard their families as being more close-knit than their white counterparts [46, 47]. This type of close-knit family relationships may facilitate grandparents' influences on black grandchildren through abovementioned pathways, exerting stronger wealth impact on child's BMI trajectories. On the other hand, it is also possible that the impact of grandparental wealth is smaller among black children than their white counterparts. Black children and families tend to face greater structural and contextual disadvantages than white children [48, 49]. For example, black families with low family wealth are more likely than their white counterparts to live in poor and disadvantaged neighborhoods [50]. Living in a deprived neighborhood may cause negative feelings among children and adolescent, such as helplessness, inferiority, and depression [51]. These negative feelings, in turn, pose important risk factors for over-eating and obesity [52, 53]. All of these consequences of living and growing up in "inferior" communities may weaken any protective function that grandparental wealth plays in influencing black grandchildren's BMI trajectories.

The goal of the present study is to assess the protective role of grandparental wealth against obesity risk in adolescence among a cohort of nationally reprehensive sample of children. We use data from the Panel Study of Income Dynamics (PSID) and its Child Development and Transition to Adulthood supplemental studies to track individuals' BMI trajectories from childhood to adolescence. We hypothesize that higher levels of grandparental wealth will be associated with more normative child BMI trajectories, while lower levels of grandparental wealth will be associated with steeper increases in BMI trajectories from early childhood to adolescence. Because the level of grandparental wealth and the environment in which black and white children grow up differ substantially and the impact of grandparental wealth on grandchildren's BMI growth may also differ, we hypothesize that the association between grandparental wealth and BMI growth trajectories can differ between white and black children.

## Data and method

Data for this analysis are from the Panel Study of Income Dynamics (PSID). Collected and managed by the Survey Research Center of the Institute for Social Research at the University of Michigan, PSID is an ongoing longitudinal survey with a nationally representative sample started with roughly 5,000 families and over 18,000 individuals in 1968. Individuals in these families and their descendants have been followed annually from 1968–1996, and biennially since 1997. In 2013, PSID collected data on 24,952 individuals (of which 17,785 are PSID "sample persons") within 9,063 families. In 1997, the PSID initiated a data collection effort for a cohort of children under the age of 13 from 2,394 families in the Child Development Supplement (CDS) [54]. The CDS was designed as a nationally representative sample of children in the United States and oversampled black and low-income families. Subsequent study waves were conducted in 2002–2003 (CDS-II) and 2007–2008 (CDS-III). When CDS participants turn 18 years of age, they become participants in the Transition to Adulthood (TA) survey. They continued to be followed repeatedly and biennially from 2005 to 2013. By 2013, the TA surveys included 2,128 young adults who were 18–27 years old. Together, these supplement surveys collect detailed information about development outcomes among children and young adults in PSID families.

Because family dynamics could be different for children living with parents and those living with grandparents or other relatives, this analysis further limited respondents to children having at least one biological parent living in the household. This is a reasonable selection

criterion because information for parental information of children who live with non-parent relatives is likely to be missing. It is important to note that only black and white children and their family members are included in the analytical sample because the number of other race children is small. We then link the CDS-TA respondents (G3) with their parents (G2) and grandparent (G1) from the PSID main survey using the family identification mapping system. The final analytic sample consisted of 1,773 G3 children/youth, of which 1,250 sample members are non-Hispanic white and 523 sample members are non-Hispanic black.

## Variables

**Body mass index.**   The heights and weights of the respondents in CDS and TA were measured by the interviewers in person repeatedly at each wave using the same brand of scale and tap. For this analysis, BMI is defined as weight/height [2]. In children aged 2 years and older, BMI provides a useful estimate of adiposity that with important health outcomes [55, 56] despite some limitations [57, 58].

**Grandparental wealth.**   The wealth measure, which the PSID collected in 1984, 1989 and 1994 (as supplements) and 1999–2013 waves of the survey, considers several financial sources: net value of the respondent's main home, other real estate, vehicles, any farms or businesses, stocks, IRAs and other financial instruments, cash accounts such as money market funds and certificates of deposit and other assets including value of estates, life insurance policies, pensions and inheritance. Any outstanding mortgage and other debts are subtracted from these assets [59]. To reduce measurement error and increase sample size, grandparents' family wealth is the average wealth between maternal and paternal grandparents. All wealth measures are adjusted to 2000-dollar values using average annual consumer price index. Grandparental wealth is calculated in two ways: net financial assets excluding or including home equity. Grandparents' wealth with equity is used as the primary wealth measure. Grandparental wealth ranking is then constructed using the wealth distribution of grandparent generation in the sample. This rank variable ranges from 0 to 1; a higher value indicates a relative advantage in wealth, and a lower value indicates a relative disadvantage in wealth. This wealth specification has the advantage of avoiding the problems associated with the skewed nature of wealth.

**Parents' SES measures.**   Similar to the way of constructing the grandparental wealth measure, parents' wealth is a time-varying measurement from 1999 to 2013. We then use wealth distribution of parent generation in the sample to construct wealth ranking. Additional socioeconomic indicators include parental education, which measured as the highest number of years of education attained by either of the two parents of the CDS child.

**Other confounders.**   We include G1's highest education in year to adjust for the potential confounders of the relationship between grandparental wealth and parental SES. We also include a series of time-varying covariates to account for the confounding between parental SES and G3 BMI, including family income, maternal BMI, number of children in the household, whether either household head is unemployed, marital status of household head. Also included are the time-invariant demographic characteristics (sex, race, and age) and low birthweight status of G3 respondents, as well as the highest educational attainment in years of either G2 parent.

## Method

In order to examine the growth trajectories of children's development, a growth curve analysis is conducted for body mass index (BMI) in the multilevel modeling context. In PSID-CDS, up to two children from the same family are randomly selected. Therefore, CDS respondents are nested within parents. Also, because PSID has a genealogical design, parents are nested within

their own parents. Given this hierarchical data structure, I follow Li [17] and use three-level growth curve models to estimate the age-based wellbeing trajectories for the CDS and TA children from 1997 to 2013. In this analysis, the repeated measurements of the grandchildren forming the first level (i.e., within-grandchildren individual effects), grandchildren forming the second level (i.e., between-grandchildren and within-grandparent effects), and grandparents (i.e., between-grandparent effects) forming the third level. The growth curve modeling strategy has several advantages. First, it allows us to examine how the intercepts (i.e. initial BMI levels) and slopes (i.e., BMI growth rate) are associated with the independent variables across individuals. Second, the model also accounts for the higher-order clusters in the data (i.e., grandchildren nested within grandparents). Finally, the model can give robust estimates from imbalanced data as not all children have the same number of observations across different time points.

Following Rabe-Hesketh, Skrondal [60], we use a fixed quadratic random linear model, which is illustrated as follows:

$$\text{Level 1}: Y = \pi_0 + \pi_1 Age + \pi_2 Age^2 + \epsilon$$

$$\text{Level 2}: \pi_0 = \beta_{00} + \sum \beta_{0q}(X_q) + \mu_0$$
$$\pi_1 = \beta_{10} + \sum \beta_{1q}(X_q) + \mu_1$$
$$\pi_2 = \beta_{20}$$

$$\text{Level 3}:$$
$$\beta_{00} = \lambda_{000} + \lambda_{001}(G1wealth) + \mu_{00}$$
$$\beta_{10} = \lambda_{100} + \lambda_{101}(G1wealth) + \mu_{10}$$

The level-1 intercept $\pi_0$, the linear slope $\pi_1$, the quadratic slope $\pi_2$, combine to describe the average trajectories in child development over time. The variance component analyses show that both the intercept $\pi_0$ and the linear slope $\pi_1$ significantly vary across individuals. Therefore, the model allows $\pi_0$ and the linear slope $\pi_1$ to vary across individuals and to be predicted by level-2 covariates $X_q$ (not including grandparents' SES, which is a level-3 covariate), where the subscript $q$ represents the $q$th of level-2 covariate.

In this analysis, we first document the changes in BMI over time by presenting the descriptive statistics. Next, we use growth curve models to predict children's BMI trajectories. Age is initiated at the minimum value to facilitate interpretation of the estimated model intercept. We then interact child age and its square term with grandparental wealth ranking to estimate the relationship between grandparents' wealth and trajectories of BMI growth. The interaction term captures age-related change in the relationship between grandparents' wealth ranking and children's BMI.

## Results

### Grandparental wealth and grandchild's body mass index trajectories

Table 1 presents the descriptive statistics of the analytical sample by child race (whites versus blacks) when the outcome variable is the repeated measure of body mass index from childhood to early adolescence. The average body mass index (BMI) for all children in the sample is around 23. On average, black children have a higher BMI in all survey years than white children. This disparity in BMI is also observed in the parent generation (33.53 versus 25.68). In addition, substantial racial differences in other independent variables are also evident. The net wealth for white children's grandparents is $274,000, compared to less than $37,000 for

**Table 1. Descriptive statistics of a three-generation sample used in the analysis of body mass index, by race: Panel Study of Income Dynamics (1968–2013), Child Development Supplement (1997–2007), and Transition to Adulthood Study (2005–2013).**

| | White grandchildren | | Black grandchildren | | All grandchildren | |
|---|---|---|---|---|---|---|
| Variable | Mean | SD | Mean | SD | Mean | SD |
| **Dependent Variables** | | | | | | |
| Body mass index[a] | 22.05 | 4.45 | 24.78 | 6.24 | 22.93 | 6.06 |
| 1997 | 17.87 | 3.85 | 19.62 | 5.53 | 18.52 | 4.68 |
| 2002 | 20.39 | 5.13 | 23.13 | 7.28 | 21.18 | 5.99 |
| 2005 | 23.25 | 3.95 | 26.67 | 5.37 | 24.46 | 4.84 |
| 2007 | 23.88 | 4.29 | 27.46 | 6.28 | 25.13 | 5.34 |
| 2009 | 24.28 | 4.55 | 27.77 | 6.65 | 25.35 | 5.44 |
| 2011 | 24.45 | 4.82 | 27.87 | 6.39 | 25.39 | 5.46 |
| 2013 | 24.63 | 5.05 | 27.24 | 6.49 | 25.33 | 5.52 |
| **Independent Variables** | | | | | | |
| G1 average net wealth (with equity, in 1000s) | 274.86 | 611.70 | 37.41 | 45.36 | 217.23 | 536.39 |
| G2 average net wealth (with equity, in 1000s) | 416.91 | 1574.26 | 95.03 | 302.03 | 302.21 | 1269.87 |
| G1 education in years | 14.20 | 3.78 | 11.46 | 3.52 | 12.17 | 5.28 |
| G2 education in years | 14.50 | 2.86 | 12.74 | 2.13 | 13.58 | 3.14 |
| G2 family income (in 1000s) | 101.54 | 96.32 | 54.58 | 22.48 | 83.62 | 104.71 |
| G2 number of kids in the household | 1.37 | 1.18 | 1.56 | 1.35 | 1.48 | 1.28 |
| G2 body mass index | 25.68 | 4.12 | 33.53 | 4.19 | 28.13 | 4.04 |
| G2 married | .90 | .29 | .72 | .38 | .87 | .32 |
| G3 baseline (1997) age | 8.45 | 2.40 | 8.61 | 2.40 | 8.46 | 2.41 |
| G3 male | .46 | .50 | .55 | .50 | .50 | .50 |
| G3 low birth weight (<2500 grams) | .05 | .22 | .12 | .33 | .08 | .25 |
| N of G3 person-periods | 3,477 | | 1,423 | | 4,900 | |
| N of G3 persons | 1,250 | | 523 | | 1,773 | |

G1, first generation (grandparents); G2, second generation (parents); G3, third generation (grandchildren); SD, standard deviation.

[a] Body mass index = weight(kg)/height(m)$^2$

grandparents of black children—that is, grandparental wealth of white children is 7.5 times higher than that of their black counterparts. This wealth disparity between black and white respondents persists in the parent generation. On average, the net parental wealth of black children is only about 20 percent that of their white peers ($95,000 versus $417,000). The average educational attainment for all respondents' grandparents and parents is 12.17 and 13.58 years, respectively. Blacks, again, are disadvantaged in this dimension of socioeconomic status: on average, the educational attainment of black respondents' grandparents and parents are about 2 to 3 years lower than that of white counterparts. Moreover, black children are disadvantaged in several other domains compared to white children. For example, relative to white children, black children have far lower family income, are more likely to be born in a lower birth weight and are more likely to grow up in single-parent families.

Table 2 shows the estimates from the growth curve models on the BMI trajectories from 1997 to 2013. The first model is an unconditional growth curve model that estimated the growth trajectories of BMI. In Model 1, the significant positive coefficients ($b = 0.98$, p<0.00) for the linear term of age suggested that BMI grows with age, and the significant negative coefficients for the quadratic term ($b = -0.03$, p<0.00) and significant cubic term ($b = 0.00$, p<0.00) of age suggested that such growth follows a convex downward-facing curvilinear pattern, with the rate of BMI growth declining with age.

**Table 2. Growth curve models of Body Mass Index (BMI) trajectories in a three-generation sample: Panel Study of Income Dynamics (1968–2013), Child Development Supplement (1997–2011), and Transition to Adulthood Study (1997–2013).**

| | Body Mass Index of Grandchild | | | | |
|---|---|---|---|---|---|
| | Model 1 | Model 2 | Model 3 | Model 4 | Model 5 |
| **Fixed Effects** | | | | | |
| *Initial BMI status* | | | | | |
| Intercept | 15.10*** | 15.65*** | 15.05*** | 14.71*** | 13.02*** |
| G1 wealth ranking | | -1.06† | -1.48* | -1.22* | 0.04 |
| G2 wealth ranking | | | 3.78** | 3.65** | 4.26*** |
| G2 BMI | | | | 0.16*** | 0.13** |
| *Linear BMI growth rate* | | | | | |
| G3 age | 0.98*** | 1.16*** | 1.24*** | 1.21*** | 1.26*** |
| G1 wealth ranking | | -0.17*** | -0.15*** | -0.13*** | -0.13*** |
| G2 wealth ranking | | | -0.40* | -0.40* | -0.46** |
| G2 BMI | | | | 0.01*** | 0.01*** |
| *Quadratic growth rate* | | | | | |
| G3 age$^2$ | -0.03*** | -0.03*** | -0.03*** | -0.03*** | -0.03*** |
| G1 wealth ranking | | 0.01 | 0.00 | 0.00 | 0.01 |
| G2 wealth ranking | | | 0.01† | 0.01* | 0.01* |
| G2 BMI | | | | 0.01** | 0.01** |
| *Cubic growth rate* | | | | | |
| G3 age$^3$ | 0.00** | 0.00** | 0.00** | 0.00** | 0.00** |
| *Control Variables* | | | | | |
| Black | | | | | 1.03 |
| Black * age | | | | | 1.65*** |
| G3 child is male | | | | | 0.17 |
| G3 low birth weight | | | | | -0.20 |
| G2 Married (at t) | | | | | 0.15 |
| G2 years of education (at 1997) | | | | | -0.01 |
| G2 family income (at t-1) | | | | | -0.14 |
| G2 number of kids in the household | | | | | -0.00 |
| G2 maternal age (at t) | | | | | -0.01 |
| **Random Effects Variance Components** | | | | | |
| Level 3: between-G1 effects | | | | | |
| Linear slope | 0.02*** | .04*** | 0.03*** | 0.03*** | 0.03*** |
| Initial status | 6.93*** | 6.10*** | 6.17*** | 5.44*** | 5.45*** |
| Cov (linear slope, initial status) | 0.99*** | 0.95*** | 0.95*** | 0.95*** | 0.95*** |
| Level 2: between-G3 and within-G1 effects | | | | | |
| Linear slope | 0.05*** | 0.03*** | 0.03*** | 0.03*** | 0.03*** |
| Initial status | 7.43*** | 6.77*** | 6.77*** | 6.56*** | 6.46*** |
| Cov (linear slope, initial status) | 0.91*** | 0.89*** | 0.89*** | 0.89*** | 0.87*** |
| Level 1: within-G3 effects | 6.14*** | 5.91*** | 5.89*** | 5.92*** | 5.68*** |
| *Goodness-of-fit* | | | | | |
| BIC | 36115.48 | 23828.93 | 23810.78 | 23660.23 | 19222.06 |
| N of G3 person-periods | 4,900 | 4,900 | 4,900 | 4,900 | 4,900 |
| N of G3 observation | 1,773 | 1,773 | 1,773 | 1,773 | 1,773 |

† p<0.1

* p<0.05

** p<0.01

*** p<0.001

Model 2 controlled for G1 wealth ranking and Model 3 additionally controlled for G2 wealth ranking. The negative coefficient for G1 wealth ranking for the model intercept indicates that the initial BMI value is lower for child of wealthier grandparents relative to child of less wealthy grandparents. More importantly, the negative and significant coefficient for G1 wealth ranking for the linear growth rate (b = -0.17, p<0.00) shows that the BMI growth rate of children of wealthier grandparents falls substantially below that of less wealthier grandparents, although the positive (yet nonsignificant) G1 wealth ranking coefficient for the quadratic growth rate implies less leveling off in wealthier than less wealthier children's BMI trajectories. Model 3 of Table 2 adds to Model 2 the wealth ranking of parent generation. The positive and large coefficient for G2 wealth ranking for the model intercept (b = 3.78, p<0.01) indicates that respondents of wealthier parents tend to have higher BMI in the initial stage than respondents of less wealthy parents. As indicated by its negative and significant coefficient for the linear growth rate and its positive and marginally significant coefficient for the quadratic growth rate, G2 wealth ranking also shapes the trajectory of child BMI over time. Individuals with wealthier parents experience slower than average growth rate as they grow up, and there is less leveling off as G2 wealth increases.

Model 3 of Table 2 additionally controlled for parent BMI. The model estimates suggest that G2 BMI is positively and significantly associated with the linear rates of child BMI growth. Moreover, the growth rate of child BMI is faster when the child's parent has higher than average BMI, and there is less leveling off as G2 BMI increases. Notably, when adding other G2- and G3-level control variables in the full model, the association between grandparental wealth ranking and the initial status of BMI is attenuated to a nonsignificant level. However, grandparental wealth ranking remains to be associated with the linear rate of their grandchild's BMI growth even with consideration of G2 wealth and BMI status.

To intuitively show how G1 wealth is associated with G3 BMI trajectory, Fig 1 plotted the predicted trajectories of child BMI of G1 wealth at the first, second, and third quartiles of these distributions. As the graph illustrated, group differences in initial BMI at age 4 are trivial. On average, children of the lowest G1 wealth have the highest initial BMI status among three groups, and they also have a faster growth rate in BMI relative to the other two groups of higher G1 wealth. In other words, children of higher G1 wealth demonstrated lower initial BMI and a slower BMI growth rate relative to child of lowest G1 wealth.

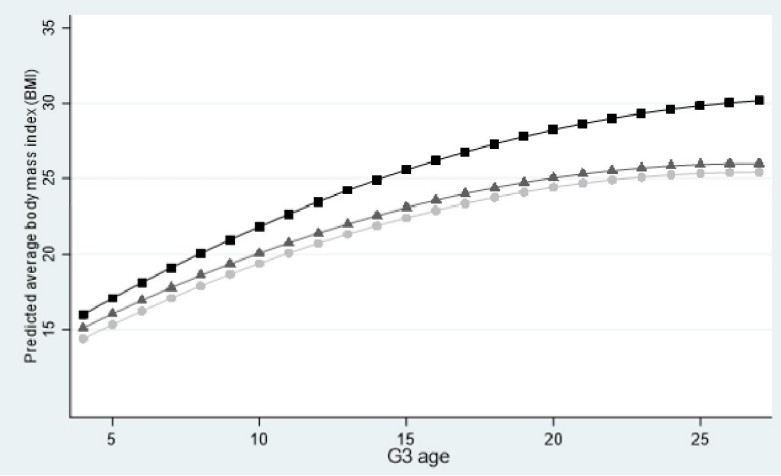

**Fig 1. Predicted trajectories of children's BMI growth trajectories, by G1 levels of wealth.** ━■━ Lowest quartile of the G1's wealth distribution. ━▲━ Middle quartiles of the G1's wealth distribution. ━●━ Top quartile of the G1's wealth distribution.

**Table 3. Growth curve models of Body Mass Index (BMI) trajectories in a three-generation sample, by race: Panel Study of Income Dynamics (1968–2013), Child Development Supplement (1997–2011), and Transition to Adulthood Study (1997–2013).**

| | White | | Black | |
|---|---|---|---|---|
| | Model 1 | Model 2 | Model 3 | Model 4 |
| **Fixed Effects** | | | | |
| Initial BMI status | | | | |
| Intercept | 14.60*** | 13.69*** | 16.52*** | 13.85*** |
| G1 wealth ranking | 1.63 | 0.70 | -0.18 | 1.28 |
| G2 wealth ranking | | 3.01* | | 4.31 |
| G2 BMI | | 0.06 | | 0.22 |
| Linear BMI growth rate | | | | |
| G3 age | 1.14*** | 1.19*** | 1.23*** | 1.41*** |
| G1 wealth ranking | -0.33* | -0.17* | -0.42 | -0.53 |
| G2 wealth ranking | | -0.37* | | -0.40 |
| G2 BMI | | 0.03* | | 0.00 |
| Quadratic growth rate | | | | |
| G3 age$^2$ | -0.03*** | -0.03*** | -0.03*** | -0.03*** |
| G1 wealth ranking | 0.01† | 0.00 | 0.01 | 0.02 |
| G2 wealth ranking | | 0.01* | | 0.01 |
| G2 BMI | | 0.00 | | -0.00 |
| *Control Variables* | | | | |
| G1 education | | -0.02 | | -0.01* |
| G3 low birth weight | | -0.61 | | -0.24 |
| G3 child is male | | 0.42* | | -0.81 |
| G2 married (at t) | | -0.48* | | 1.06* |
| G2 years of education (at 1997) | | 0.01 | | -0.05 |
| G2 family income (at t-1) | | -0.00 | | 0.00 |
| G2 maternal age (at t) | | -0.03 | | -0.02 |
| **Random Effects Variance Components** | | | | |
| Level 3: between-G1 effects | | | | |
| Linear slope | 0.03** | 0.03** | 0.01*** | 0.00*** |
| Initial status | 2.62*** | 2.20*** | 12.62*** | 12.54*** |
| Cov (linear slope, initial status) | 0.05*** | 0.03*** | -0.14*** | -0.22*** |
| Level 2: between-G3 and within-G1 effects | | | | |
| Linear slope | 0.02** | 0.02** | 0.07** | 0.07** |
| Initial status | 5.80*** | 5.66*** | 9.98*** | 10.86*** |
| Cov (linear slope, initial status) | -0.11*** | -0.12*** | -0.11*** | -0.24*** |
| Level 1: within-G3 effects | 4.92*** | 4.84*** | 10.46*** | 10.37*** |
| Goodness-of-fit | | | | |
| BIC | 16730.56 | 16473.71 | 5856.965 | 5578.949 |
| *N* of person-periods | 3,477 | 3,477 | 1,423 | 1,423 |
| *N* of persons | 1,250 | 1,250 | 523 | 523 |

† p<0.1

\* p<0.05

** p<0.01

*** p<0.001

To test whether results differ between black and white children, we fit a series of parallel growth curve models separately for white and black children. Table 3 shows the estimates from

these race-specific models. The first model for each race group (Model 1 for whites and Model 3 for blacks) controls for G1 wealth, while the second model of each race (Model 2 for whites and Model 4 for blacks) additionally controls for G2 SES factors and G2 BMI, as well as other covariates. Across all models, the significant positive coefficients for the linear term of age suggest that BMI grows with age, and the significant negative coefficients for the quadratic term of age suggests that the rate of BMI increase declines with age. When the coefficients of linear and quadratic growth rate are compared, the BMI growth rate is faster for black children than that of white children. Grandparents' wealth is not significantly associated with the initial status of BMI trajectories for both groups. However, grandparents' wealth is significantly and negatively associated with the linear growth rate of BMI for white children but not for black children. For whites, the linear rate of BMI growth would have decreased if the grandchild had a wealthier grandparent. As indicated in Models 2 and 4, G2 wealth and BMI are significantly associated with the rate of BMI growth for white children but not for black children. Adding these G2 and G3 variables also attenuated the coefficients of G1 wealth considerably, indicating that a large part of the growth trajectories of BMI growth is shaped by G2 wealth and BMI status. Consistent with previous studies, maternal BMI is a strong predictor of child's BMI and a major factor in the intergenerational transfer of body weight status throughout childhood and adolescent years. [61]

In supplementary analysis, alternative measurements of wealth were used to test if the results are robust to the coding of grandparental wealth. First, we estimated models with a categorical measure of grandparental wealth (lowest quartile of G1 wealth, middle two quartiles of G1 wealth, and top quartile of G1 wealth). We also used grandparental wealth ranking without home equity as an alternative dependent variable. The supplemental results suggest that the relationships between grandparental wealth and the growth trajectory of child BMI report here are robust to alternative G1 wealth measures.

## Discussion

This study explored how grandparents' wealth is related to child obesity risk, measured by child BMI growth trajectories from early childhood to adolescent years. In our study, grandparental wealth ranking was a significant predictor of BMI trajectories such that higher wealth ranking is associated with slower BMI growth rate. Our findings provide some evidence that exposure to grandparental wealth had a long-lasting impact on obesity risk trajectories.

Results show that compared to children with more grandparental wealth, children with less grandparental wealth demonstrated a higher initial BMI, and a faster rate of BMI growth from early childhood through adolescence. The association persists even after controlling for a variety of parental sociodemographic characteristics. It indicates that the lack of grandparental wealth may put children at risk of excessive weight gain and obesity. This is perhaps because children of low grandparental wealth are more likely to have repeated exposure to energy-dense food and limited access to fruit and vegetables [62, 63] and tend to develop unhealthy food preferences [64]. Furthermore, early childhood is an important period for individuals to develop habits of physical activity that will carry on lifelong. Having limited grandparental wealth may prevent children from engaging in an active lifestyle, which in turn, may impair their development of healthy physical activity habits as they grow up [65, 66]. All of these may put children of low grandparental wealth on a trajectory of faster weight gain.

The analysis also suggests that the association between grandparental wealth and grandchildren's BMI growth rate differs by children's race/ethnicity. Grandparental wealth of white children plays a significant role in shaping grandchildren's early development in BMI, and the effect of wealth diminishes as children grow and develop. It may suggest that an initial

favorable body weight feeds into later favorable development among white children [67, 68]. For black children, the association between grandparental wealth ranking and BMI growth trajectories is not statistically significant. Several studies have also found a weak or non-existent association between SES and other health outcomes among African Americans [69, 70]. A few explanations may account for the lack of association between grandparental wealth and the obesity risk among black children. First, the sample size of black children and their intergenerational linkages are limited, and the variation in their wealth distribution is also small. Thus, the non-significant association may result from a limited statistical power. Second, it is possible that racism and discrimination, as well as disadvantaged social, economic, and environmental conditions, lessen and even offset the potential beneficial effects of grandparental wealth. Consequently, higher levels of grandparental wealth may be less readily converted to improved BMI growth trajectories among black children. Finally, the weaker influence of grandparental SES for black children may be attributable to the disrupted intergenerational transmission processes. For instance, the incarceration rate among blacks is more than eight times higher than those of whites [71], and the marital dissolution rate among black families is much higher than white families [72]. These severe types of family disruptions may weaken the grandparental role in grandchildren's lives, leading to weaker impact of grandparental wealth on grandchildren's BMI growth trajectories.

The analysis is not without limitations. First, the analysis does not investigate the mechanisms concerning *how* grandparental wealth matter for grandchildren's BMI trajectories. Given the strong link between built-in environment and children's body mass index [73], future research may use restricted PSID data to investigate whether the effect of grandparental wealth operates through help provided by grandparents to help adult children and their grandchildren to move away from a poor neighborhood to a better-off community. Doing so holds promise for developing effective intervention programs that reduce child obesity and the racial disparities therein. Furthermore, recent research suggests that epigenetic factors, such as DNA methylation, play a role in the biological mechanisms through which social factors are linked to obesity risk. Although our analysis controlled for maternal BMI, we are unable to assess the relevance of the biological mechanism due to data limitations. Future research might utilize alternative datasets that provide epigenetic markers to investigate whether the association between grandparental wealth and children's BMI trajectories persists after accounting for the biological mechanism.

In addition, the analytical data are structured hierarchically and only include individuals in the grandparent generation who have grandchildren. Selecting grandchildren based on the availability of prospective grandparental information (of wealth) might also create a sample very distinct from the general population because of the multigenerational survival and fertility patterns required to make three generations available to participate in the PSID during its four decades of follow-up. Somewhat relatedly, grandparental wealth is measured as the average of maternal and paternal grandparents' family wealth. The impact of grandparental wealth on child wellbeing may differ between paternal and maternal linkages [74]. Therefore, efforts should be devoted to exploring whether and how the protective role of wealth might vary between paternal and maternal grandparents.

Finally, the analysis is limited to a three-generation analysis largely due to data limitation. However, it is more than likely that the wellbeing of current generation is influenced by the wealth status of previous generations over and above that of parents [75]. Some economists estimate that half or more of life time family wealth accumulation can be attributed to past generations in the form of gifts, inheritances, or indirect support [76, 77]. Therefore, wealth may have a particularly long arm in influencing future generations' wellbeing. Future research might benefit from a closer examination of whether and how children's wellbeing in general is linked to the historical socioeconomic advantages and disadvantages of previous generations.

## Conclusion

In conclusion, results from this study indicated that grandparental wealth has enduring protective effects on altering obesity risk from early childhood to adolescents. The effects seem to be more pronounced for white children than for black children. Public policies promoting wealth accumulation of families might be a new and promising avenue for the prevention of obesity risks of the next generations. In addition, government programs and interventions targeted at mitigating obesity risk would likely be beneficial in considering family wealth distribution and children's racial and ethnic backgrounds [78].

## Author Contributions

**Conceptualization:** Ying Huang.

**Writing – original draft:** Ying Huang.

**Writing – review & editing:** Ying Huang.

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
