## [Decision Letter · Decision Letter 0]

27 Jan 2020

PONE-D-19-32075

Grandparents’ Wealth and the Body Mass Index Trajectories of Grandchildren

PLOS ONE

Dear Dr. Huang,

Thank you for submitting your manuscript to PLOS ONE. After careful consideration, we feel that it has merit but does not fully meet PLOS ONE’s publication criteria as it currently stands. Therefore, we invite you to submit a revised version of the manuscript that addresses the points raised during the review process.

We would appreciate receiving your revised manuscript by Mar 12 2020 11:59PM. To enhance the reproducibility of your results, we recommend that if applicable you deposit your laboratory protocols in protocols.io, where a protocol can be assigned its own identifier (DOI) such that it can be cited independently in the future. For instructions see: http://journals.plos.org/plosone/s/submission-guidelines#loc-laboratory-protocols

We look forward to receiving your revised manuscript.

Kind regards,

Tabassum Insaf

Academic Editor

PLOS ONE

Reviewers' comments:

Reviewer's Responses to Questions

**Comments to the Author**

1. Is the manuscript technically sound, and do the data support the conclusions?

Reviewer #1: Yes

Reviewer #2: Yes

2. Has the statistical analysis been performed appropriately and rigorously? 

Reviewer #1: Yes

Reviewer #2: Yes

3. Have the authors made all data underlying the findings in their manuscript fully available?

Reviewer #1: Yes

Reviewer #2: Yes

4. Is the manuscript presented in an intelligible fashion and written in standard English?

Reviewer #1: Yes

Reviewer #2: Yes

5. Review Comments to the Author

Reviewer #1: This is an impressive study showing convincingly that grandparent wealth is associated with grandchild BMI trajectory. For the most part, the research questions are well-justified, the analyses are rigorous, and the conclusions are valid. However, I also offer the following critiques intended to improve the manuscript:

1) First, the Introduction could be improved with a more focused discussion of why we might expect to see race differences in the association between grandparent wealth and grandchild BMI. The existence of a race-based wealth gap is clear (and cited in the manuscript), as is the existence of race-based differences in obesity risk. However, the Introduction does not seem to include a focused argument for why the ASSOCIATION between wealth and BMI may be different in African American vs. White families.

2) The discussion of the results of the analyses of race differences could also benefit from further development. While it is true that the relevant estimates for the White families tend to be statistically significant, and the estimates for the Black families tend to be non-significant, the actual point estimates are larger for the Blacks than the Whites. This detail, I think, should be noted in the manuscript. Given this pattern of results, I do not know that it is reasonable to conclude so starkly that grandparent wealth plays a "vital" role for White, but does "not seem to matter" for Blacks.

3) I would suggest avoiding the use of the terms "accelerated" or "decelerated" to describe inter-individual differences in BMI trajectories. An intra-individual trajectory may accelerate or decelerate, but when comparing one person's (or group's) trajectory to another's, these terms do not apply well. So, for example, in the Abstract, I would suggest editing the Results section to read: "Further, increases in grandparental wealth in childhood are associated with a SLOWER BMI growth rate."

Reviewer #2: The paper uses the Panel Study of Income Dynamics (PSID) merged with the Child Development Supplement (CDS) and Transition to Adulthood (TA) Supplement to examine the relationship between parental (G2) and grandparental (G1) wealth in early childhood and children’s BMI. The paper finds that additional grandparental wealth leads to lower BMI, mainly for white children. This paper represents a useful example of the impacts of multi-generations on child well-being. While there are many studies demonstrating the importance of parent SES, there are only a few studies showing the benefits of grandparents.

The paper could be improved by providing additional details on the relationship between G1 wealth and G2 income and wealth. All of these are correlated and could be influencing the regression results. While the paper creates wealth rankings, it uses the parental income levels directly. The rankings could be masking the importance of large wealth levels. Households in the bottom two quintiles of wealth have very little wealth (the bottom quintile even has zero or negative net worth). In addition, it would be useful to present the correlation between G1 and G2 wealth (and rankings), as well as between G2 income and wealth. It could be that the G1 results are driven by the households who have both high G1 and G2 wealth. The authors could consider including a G1*G2 term, or use a categorical variable for G1 and G2 wealth quartile indicators and their cross-terms.

The authors should also explain the troubling result that parental wealth actually increases the initial BMI. In addition, it is counterintuitive that the paper finds that G1 wealth has a larger impact on reducing BMI than G2 wealth. This could all be due to the correlation between G1 and G2 wealth.

Finally, the results in Table 3 suggest that all of the effects are accruing to white children. Again, this could be due to the distribution of wealth. As Table 1 shows, the distribution of G1 wealth among white children is much more disperse than among black children (CV of 2.2 for whites compared to 1.2 for blacks). The authors could consider including black and white interaction terms for the age squared terms in the model in Table 2. It could be that all of the action will be in the white interaction terms.

However, the authors need to spend some time discussing the mechanisms for why G1 wealth impacts the BMI for white children and not black children. The authors suggest that this could be due to other factors such as neighborhood effects. The authors could examine this by using the restricted access PSID data with geocodes.

One issue concerns the data set; do all children have grandparents and how does that impact the results for those children without living grandparents or without grandparents, and are these the maternal or paternal grandparents or only those who are in the survey. The latter could also lead to some selection

6. PLOS authors have the option to publish the peer review history of their article (what does this mean?). If published, this will include your full peer review and any attached files.

Reviewer #1: No

Reviewer #2: No

---

## [Author Response · Author response to Decision Letter 0]

14 Feb 2020

Reviewer 1

1. First, the Introduction could be improved with a more focused discussion of why we might expect to see race differences in the association between grandparent wealth and grandchild BMI. The existence of a race-based wealth gap is clear (and cited in the manuscript), as is the existence of race-based differences in obesity risk. However, the Introduction does not seem to include a focused argument for why the ASSOCIATION between wealth and BMI may be different in African American vs. White families.

We agree with the reviewer that our initial discussion on the possible racial differences in the association between grandparental wealth and grandchildren’s BMI was inadequate. Following the reviewer’s suggestion, we now expand our discussion on why grandparental wealth may have differential impact on black and white grandchildren’s BMI growth trajectories, respectively. We note that the wealth-BMI association is stronger among blacks than whites, given the close-knit family relationships among black families may facilitate and enhance the protective function of grandparental wealth on grandchildren’s BMI growth trajectories. It is also possible that the wealth-BMI association is weaker among blacks than among whites because the social and contextual disadvantages that black children are exposed to may offset or weaken the protection function of grandparental wealth. This addition can be found on page 5.

2. The discussion of the results of the analyses of race differences could also benefit from further development. While it is true that the relevant estimates for the White families tend to be statistically significant, and the estimates for the Black families tend to be non-significant, the actual point estimates are larger for the Blacks than the Whites. This detail, I think, should be noted in the manuscript. Given this pattern of results, I do not know that it is reasonable to conclude so starkly that grandparent wealth plays a "vital" role for White, but does "not seem to matter" for Blacks.

We agree with the reviewer that the results of the racial differences in the associations between grandparental wealth and grandchildren’s BMI trajectories need further development. Following the reviewer’s suggestion, we add a sentence noting that despite the nonsignificant associations, the impact of grandparental wealth seems to be larger than their white peers based on the point estimates of the coefficients. We also suspect that the non-significant results among African American child sample may result from a limited statistical power, and call for future research to explore this issue further. This addition can be found on pages 15-16. 

We agree with the reviewer that it is not appropriate to conclude that grandparental wealth does not seem to matter for African American children. We delete the original wording, and note that the social and contextual disadvantages experienced by African American children and their families (among other possible reasons) may have weakened (or offset) the protective role of grandparental wealth in mitigating the obesity risk. This change can be found on page 16. 

3. I would suggest avoiding the use of the terms "accelerated" or "decelerated" to describe inter-individual differences in BMI trajectories. An intra-individual trajectory may accelerate or decelerate, but when comparing one person's (or group's) trajectory to another's, these terms do not apply well. So, for example, in the Abstract, I would suggest editing the Results section to read: "Further, increases in grandparental wealth in childhood are associated with a SLOWER BMI growth rate."

Response: We thank the reviewer for the careful read. Following the reviewer’s suggestion, we now replace the term “accelerated” and “decelerated” with “faster” and “slower” throughout the manuscript. 

Reviewer 2: 

1a). The paper could be improved by providing additional details on the relationship between G1 wealth and G2 income and wealth. All of these are correlated and could be influencing the regression results. While the paper creates wealth rankings, it uses the parental income levels directly. The rankings could be masking the importance of large wealth levels. Households in the bottom two quintiles of wealth have very little wealth (the bottom quintile even has zero or negative net worth). 

Response: The reviewer is correct that socioeconomic status indicators of the parent and grandparent generations are correlated to each other. For example, the correlation between G1 and G2 wealth ranking is .38 for all respondents. The correlation between G2 wealth and G2 income is .42 all respondents. In both instances, the correlations are higher for whites than for blacks. In supplemental analysis, we run correlation matrix among all independent variables. None of the variance inflation factor (VIF) values exceed 10, indicating that our models do not have multicollinearity issue. 

As the reviewer surmises, the wealth distribution is skewed, and it has zero and negative values. A log transformation does not allow zero or negative values, which are common in wealth measures. To avoid the skewness of the wealth distribution, and also to retain the zero and negative wealth values, we opt to use a wealth ranking constructed from a cumulative distribution function that is bounded by 0 and 1 for the parent and grandparent generation, respectively. 

In addition to the wealth ranking measurement, we also use a categorical specification of grandparental wealth in the supplemental analysis, where we categorize wealth into three levels: lowest quartile of G1 wealth, middle two quartiles of G1 wealth, and top quartile of G1 wealth. As mentioned in the supplemental analysis section on page 14, results from this additional analysis are qualitatively similar to the results report in the manuscript.

b). In addition, it would be useful to present the correlation between G1 and G2 wealth (and rankings), as well as between G2 income and wealth. It could be that the G1 results are driven by the households who have both high G1 and G2 wealth. The authors could consider including a G1*G2 term, or use a categorical variable for G1 and G2 wealth quartile indicators and their cross-terms.

As suggested by the reviewer, we considered presenting a correlation matrix table to describe the correlation between variables used in the analysis. In the end, however, we feel that this proposed correlation matrix table introduced repetition. For this reason, we have decided not to present information regarding the correlations between socioeconomic measures. To answer this reviewer’s query, and as mentioned in our response to this reviewer’s point #1, the correlation between G1 and G2 wealth ranking is .38 for all respondents. The correlation between G2 wealth and G2 income is .42 for all respondents. 

We agree with the reviewer that it is valuable to test the whether the effect of grandparental wealth on grandchildren varies by the levels of socioeconomic resources that the parent generation possesses. However, we feel this additional analysis may divert the attention from the primary goal of the paper. Nevertheless, in supplemental analysis, we test two competing hypotheses, as suggested by the reviewer: 1) augmentation hypothesis, which posits the impact of grandparental wealth on child’s BMI growth trajectories is stronger for children of high SES parents than children of low SES parents, and 2) compensation hypothesis, which predicts that the impact of grandparental wealth on child’s BMI growth trajectories is stronger for children of low SES parents than for children of high SES parents. We find that the impact of grandparental wealth on children’s BMI growth trajectories is stronger for children of high SES families than children of low SES families, providing tentative support to the augmentation hypothesis. The results suggest that there may be some value in exploring how the impact of grandparental wealth may vary based on the SES levels of the parent generation. We leave this important task for future research. 

2. a). The authors should also explain the troubling result that parental wealth actually increases the initial BMI. 

In supplemental analysis, we estimate the association between G2 wealth and child BMI growth trajectories. We find that, even without the influence of G1 wealth, G2 wealth is still positively and significantly associated with child’s initial BMI levels (intercept=2.56, p<0.05).

We note that the positive and statistically significant parameter estimate for the intercept of the G2 wealth ranking (3.78) shows that children whose parental wealth ranking is high tend to have higher initial BMI levels than their peers with lower parental wealth at very young ages. However, this positive association levels off at an accelerated rate (as indicated by the negative linear growth rate and positive quadratic growth rate). This may indicate that parental wealth—a reflection of positive home environment and optimal nutritional status—in early childhood (around age 5) is positively associated with higher BMI levels at the baseline. 

b). In addition, it is counterintuitive that the paper finds that G1 wealth has a larger impact on reducing BMI than G2 wealth. This could all be due to the correlation between G1 and G2 wealth.

In fact, Table 2 and 3 show that the linear growth rates for G1 wealth are consistently smaller than the linear growth rates for G2 wealth, indicating G2 wealth has a greater impact than G1 wealth on reducing children’s BMI growth rate. Although this result might result from a correlation between G1 and G2 wealth, we note that G1 and G2 wealth are not highly correlated to each other. We feel it is important to control for G2 wealth when examining the net effect of grandparental wealth on grandchildren’s BMI trajectories. 

3. Finally, the results in Table 3 suggest that all of the effects are accruing to white children. Again, this could be due to the distribution of wealth. As Table 1 shows, the distribution of G1 wealth among white children is much more disperse than among black children (CV of 2.2 for whites compared to 1.2 for blacks). The authors could consider including black and white interaction terms for the age squared terms in the model in Table 2. It could be that all of the action will be in the white interaction terms.

As mentioned in our response to Reviewer’s #1’s point 2, we now expand our discussion on the non-significant impact of grandparental wealth on black children’s BMI growth trajectories. We note that statistical power, as the reviewer suggests, could be one of the explanations for the non-significant findings. We also note that several studies have also found a weak or non-existent association between SES and other health outcomes among African Americans. It is possible that racism and discrimination, or the chronic stress induced by racism/discrimination, might lessen the potential beneficial effects of grandparental wealth. A related explanation is that residential racial segregation results in worse-off social, economic and environmental contextual conditions even for African Americans with relatively high grandparental wealth. Consequently, higher levels of grandparental wealth may be less readily converted to improved BMI growth trajectories. This material can be found on page 16.

4. However, the authors need to spend some time discussing the mechanisms for why G1 wealth impacts the BMI for white children and not black children. The authors suggest that this could be due to other factors such as neighborhood effects. The authors could examine this by using the restricted access PSID data with geocodes.

We agree with the reviewer completely that the restricted PSID data, which offer geographic information of respondents, will be one of the ideal datasets to disentangle the mechanisms through which grandparental wealth positively influence child wellbeing. We do not have access to the restricted PSID data at this point, and we call for future research to address this study limitation. This addition can be found on page 16. 

5. One issue concerns the data set; do all children have grandparents and how does that impact the results for those children without living grandparents or without grandparents, and are these the maternal or paternal grandparents or only those who are in the survey. The latter could also lead to some selection.

The reviewer is correct that sample selection (and attrition) is a concern. We note that the analytic sample may be restrictive due to the increased chance of selection bias resulting from mortality and fertility. The analytical data are structured hierarchically and only include individuals in the grandparent generation who have grandchildren. However, processes such as assertive mating and fertility may be related to the socioeconomic status of grandparents. Selecting grandchildren based on the availability of prospective grandparent information (of wealth) might also create a sample that is distinct from the general population because of the multigenerational survival and fertility patterns required to make three generations available to participate in the PSID during its four decades of follow-up. We call for future research to use a representative G2 cohort and add information about the parents of members of the cohort (i.e., G1 characteristics) as necessary and follow the cohort forward to add offspring information (i.e., G3 characteristics) for those who become parents. Finally, one of the general limitations in almost all longitudinal study is attrition (Fitzgerald 2011). Studies on the PSID data suggest that attrition of children born to the PSID sample is not random but is higher among the socioeconomically disadvantaged and racial minorities(Fitzgerald, Gottschalk and Moffitt 1998). Nevertheless, attrition in PSID has little impact on parameter estimates in models relating family SES background to children’s outcomes (Fitzgerald 2011, Fitzgerald, Gottschalk and Moffitt 1998). Moreover, the attrition only suggests that the findings are conservative. Given that children of racial minorities are disproportionally represented in the attrited sample and previous literature generally suggests that the minorities tend to have worse health and lower cognitive skills, analyses excluding these individuals may produce underestimated coefficients. We added some materials around sample selection and its implications on page 17. 

References:

Fitzgerald, J., P. Gottschalk, and R. Moffitt. 1998. "An Analysis of Sample Attrition in Panel Data: The Michigan Panel Study of Income Dynamics." The Journal of Human Resources 33(2):251-299.

Fitzgerald, J.M. 2011. "Attrition in Models of Intergenerational Links Using the Psid with Extensions to Health and to Sibling Models." B E J Econom Anal Policy 11(3).

---

## [Decision Letter · Decision Letter 1]

16 Apr 2020

Grandparents’ Wealth and the Body Mass Index Trajectories of Grandchildren

PONE-D-19-32075R1

Dear Dr. Huang,

We are pleased to inform you that your manuscript has been judged scientifically suitable for publication and will be formally accepted for publication once it complies with all outstanding technical requirements.

With kind regards,

Man Ki Kwok

Academic Editor

PLOS ONE

Additional Editor Comments (optional):

Reviewers' comments:

Reviewer's Responses to Questions

**Comments to the Author**

1. If the authors have adequately addressed your comments raised in a previous round of review and you feel that this manuscript is now acceptable for publication, you may indicate that here to bypass the “Comments to the Author” section, enter your conflict of interest statement in the “Confidential to Editor” section, and submit your "Accept" recommendation.

Reviewer #1: All comments have been addressed

Reviewer #2: All comments have been addressed

2. Is the manuscript technically sound, and do the data support the conclusions?

Reviewer #1: Yes

Reviewer #2: Yes

3. Has the statistical analysis been performed appropriately and rigorously? 

Reviewer #1: Yes

Reviewer #2: Yes

4. Have the authors made all data underlying the findings in their manuscript fully available?

Reviewer #1: Yes

Reviewer #2: Yes

5. Is the manuscript presented in an intelligible fashion and written in standard English?

Reviewer #1: Yes

Reviewer #2: Yes

6. Review Comments to the Author

Reviewer #1: The authors have addressed my previous comments. I have no further comments.

Reviewer #2: (No Response)

7. PLOS authors have the option to publish the peer review history of their article (what does this mean?). If published, this will include your full peer review and any attached files.

Reviewer #1: No

Reviewer #2: No

---

## [Editor Report · Acceptance letter]

21 Apr 2020

PONE-D-19-32075R1 

Grandparents’ Wealth and the Body Mass Index Trajectories of Grandchildren 

Dear Dr. Huang:

I am pleased to inform you that your manuscript has been deemed suitable for publication in PLOS ONE. Congratulations! Your manuscript is now with our production department. 

With kind regards,

on behalf of

Dr. Man Ki Kwok 

Academic Editor

PLOS ONE